# Chinese Cordyceps: Bioactive Components, Antitumor Effects and Underlying Mechanism—A Review

**DOI:** 10.3390/molecules27196576

**Published:** 2022-10-04

**Authors:** Yan Liu, Zhi-Jian Guo, Xuan-Wei Zhou

**Affiliations:** School of Agriculture and Biology, Engineering Research Center of Therapeutic Antibody, Shanghai Jiao Tong University, Shanghai 200240, China

**Keywords:** Chinese Cordyceps, antitumor, immunomodulatory, bioactive components, mechanism

## Abstract

Chinese Cordyceps is a valuable source of natural products with various therapeutic effects. It is rich in various active components, of which adenosine, cordycepin and polysaccharides have been confirmed with significant immunomodulatory and antitumor functions. However, the underlying antitumor mechanism remains poorly understood. In this review, we summarized and analyzed the chemical characteristics of the main components and their pharmacological effects and mechanism on immunomodulatory and antitumor functions. The analysis revealed that Chinese Cordyceps promotes immune cells’ antitumor function by via upregulating immune responses and downregulating immunosuppression in the tumor microenvironment and resetting the immune cells’ phenotype. Moreover, Chinese Cordyceps can inhibit the growth and metastasis of tumor cells by death (including apoptosis and autophagy) induction, cell-cycle arrest, and angiogenesis inhibition. Recent evidence has revealed that the signal pathways of mitogen-activated protein kinases (MAPKs), nuclear factor kappaB (NF-κB), cysteine–aspartic proteases (caspases) and serine/threonine kinase Akt were involved in the antitumor mechanisms. In conclusion, Chinese Cordyceps, one type of magic mushroom, can be potentially developed as immunomodulator and anticancer therapeutic agents.

## 1. Bioactive Components

According to the epidemiological studies, a high intake of foods rich in bioactive compounds is beneficial to human health. Well-known as medicinal mushrooms, both natural and cultured products of Chinese Cordyceps many bioactive components. Over 20 bioactive ingredients have been isolated and their bioactivities/pharmacological effects have been proven (Table 1), of which, adenosine, cordycepin and polysaccharides are of significance in antitumor and immune regulation and considered to be the most important properties of Chinese Cordyceps [1,2]. In addition, Chinese Cordyceps is rich in fatty acids (including saturated and unsaturated fatty acids), vitamins, metal elements and other components, which also have positive effects on reducing blood lipids, preventing cardiovascular diseases, protecting kidneys and improving the essence of life [3,4].

### 1.1. Adenosine and Cordycepin

Adenosine is regarded as an important marker reflecting the quality of Chinese Cordyceps in Chinese Pharmacopoeia [84,85]. The molecular weight (MW) of cordycepin is 267.245 Da, molecular formula C_10_H_13_N_5_O_4_ and structure is shown in Figure 1A. The content of adenosine in Chinese Cordyceps ranges from 0.28 to 14.15 mg/g [86]. Cordycepin, another important nucleoside, is a purine alkaloid and confirmed as 3′-deoxyadenosine, similar in structure to adenosine except for lacking a hydroxyl group in the 3′ position of its ribose moiety (Figure 1B). The MW of cordycepin is 251.24 Da, molecular formula C_10_H_13_N_5_O_3_. The content of cordycepin ranges from 0.006 to 6.36 mg/g in *O. sinensis* [87] and can reach 2.28 mg/g in cultured *C. militaris* [88].

Both adenosine and cordycepin belong to nucleosides, and nucleosides can be dissolved with water, methanol and ethanol [84], while they are insoluble in benzene, ether or chloroform. Usually, Chinese Cordyceps is ground into powder and extracted with methanol aqueous solution or distilled water using an ultrasonic machine or a soxhlet extractor to obtain nucleosides [80,89]. Additionally, phosphate-buffered saline (PBS) is used as the solvent.

### 1.2. Polysaccharides

Polysaccharides are another important biologically active component of Chinese Cordyceps, with a content of 3–8% of the total weight [47]. The structure of polysaccharides is currently inconclusive, and a large amount of work was performed to try to explore its structure characteristic. It is reported that medicinal fungal polysaccharides with immunomodulatory and antitumor effects are most common glucans linked by various glycosidic bonds, such as (1→3)-, (1→6)-β, or (1→3)-, (1→4)-, (1→6)-α-glucans [90], and heteropolysaccharides are very important bioactive polysaccharides in Chinese Cordyceps [91]. Figure 1C shows the common structures of Chinese Cordyceps polysaccharides. Moreover, the pharmacological activity of polysaccharides is corelated to its MW. The larger the MW (10–1000 kDa) of the polysaccharides, the greater the water solubility and the better the biological activity [3]. It is reported that fungi polysaccharides have antitumor activity only when the MW is greater than 16,000 Da [47]. Usually, polysaccharides are colorless and odorless, and stably dissolved in water [92]. Polysaccharides have good water solubility, and water extraction combined with alcohol precipitation is a very effective method to extract active polysaccharides. Cordyceps polysaccharides include two forms: extracellular polysaccharide (EPS), mainly sourced from the fermentation broth of submerged *Cordyceps* spp., and intracellular polysaccharide (IPS), mainly resourcing from the fruiting bodies of Chinses Cordyceps and cultured mycelium. The chemical structure and immunomodulatory and antitumor activities of Polysaccharides originated from *Cordyceps* spp. are shown in Table 2.

## 2. Antitumor Activities and Their Participation in Molecular Mechanisms

### 2.1. Enhancing Antitumor Immune Responses

Studies have shown that the occurrence and development of tumors are closely related to immune surveillance. Immunotherapy has been proved to be an effective method to treat a variety of cancers [99] and increasing data has shown that antitumor effects of enhancing immunity may be associated with its action for the regulation of the tumor immune environment [100,101]. Immune cells show different phenotypes in response to various environmental cues (microbial products, damaged cells, cytokines, etc.). Chinese Cordyceps has a biphasic regulatory effect on the immune-cell phenotype and can increase antitumor immune activity in the tumor immune microenvironment (TIM): increasing the proinflammatory phenotypic while reversing the suppressive phenotype (Table 3). 

Chinese Cordyceps as an immunomodulator has suppressive effects on the immune system. Cordycepin has demonstrated to inhibit the differentiation of T cells into regulatory T cells (Treg, a suppressive phenotype of T cells) and delay tumor growth in tumor-bearing mice [39]. The further investigation reveals that cordycepin decreased the secretions of interleukin-2 (IL-2) and transforming growth factor-β (TGF-β) which were essential for Treg cells’ proliferation and differentiation. In addition, macrophages are the key player in the immune system which can engulf and destroy foreign pathogens and cancer cells. Tumor-associated macrophages (TAM), taking up 50% of the infiltrated cells at the tumor site, can be differentiated into two phenotypes: M1 phenotype (classic activation polarization) or M2 phenotype (alternative activation polarization), based on the stimulatory signals from the tumor microenvironment (TME) [104]. Macrophages in the TME are predominantly in an M2 state [105], and currently M2 macrophages are potential targets for the treatment of cancer. Activated M2 macrophages would suppress the immune system and promote tumor progression by releasing immunosuppressive cytokines (i.e., IL-4 and IL-10) and recruiting Th2 and Treg cells [106,107]. Clinical studies have proposed a new strategy where reversing M2 into the M1 phenotype is an effective approach to enhance antitumor immunity [108,109]. Chinese Cordyceps are capable to of resetting the macrophage phenotype repolarizing M2 to M1 macrophages. A study by Chen et al. [103] showed that APSF, a polysaccharide isolated from the fruiting bodies of *O. sinensis*, reversed M2 to the M1 phenotype through reducing the expression of IL-10 and increasing the expression of tumor necrosis factor-alpha (TNF-α), IL-12 and inducible nitric oxide synthase (iNOS), and downregulating the expressions of SR and MR (Scavenger Receptor and Mannose Receptor, M2 markers), in Ana-1 mouse macrophages co-cultured with a supernatant of H22 cells. Additionally, a novel polysaccharide CMPB90-1 from *C**. militaris* was found to remodel TAMs from M2 to the M1 phenotype through decreasing the mRNA expression level of immunosuppressive cytokines (IL-10, TGF-β and Arg-1 (arginase 1), M2 markers) while increasing the mRNA expression levels of IL-12 and iNOS (M1 markers). Additionally, a further investigation revealed that the signaling pathways of p38, extracellular-signal-regulated kinases (ERK), Akt and nuclear factor kappaB (NF-κB) were activated [48]. These findings demonstrated that Chinese Cordyceps and its bioactive constituents could promote immune cells’ antitumor function by enhancing immune responses and downregulating immune suppression (Figure 2).

### 2.2. Direct Antitumor Effects

To date, increasing studies have shown that Chinese Cordyceps has significant antitumor effects. Although the mechanisms of action are complicated, the possible mechanisms of antitumor action of Chinese Cordyceps are summarized and presented in Table 4.

#### 2.2.1. Inducing Apoptosis and Autophagy

Apoptosis is a form of programmed cell death and essential for the development and homeostasis of organisms, and its abnormal regulation is perhaps related to tumor formulation. Inducing apoptosis involves two major pathways: the intrinsic pathway (particularly mitochondrial stress) and extrinsic signal pathway. The Fas/FasL system plays an important role in apoptosis regulation. Fas and its ligand FasL are mainly expressed on the cell membrane surface. When external FasL is expressed by cytotoxic T lymphocytes and combines with Fas which is expressed by target cells, Fas-associated death domain (FADD) will be formed. FADD triggers apoptosis through recruiting extrinsic stimuli and death receptors (DRs) [125,126]. A study by Lee et al. [127] showed that cordycepin inhibited proliferation and induced HT-29 colon cancer cells’ apoptosis by increasing expression of DR3, caspase-8, -1 and -3. Caspases is a family of cysteine proteases and acts on proteins or enzymes related to the cytoskeleton or DNA and is responsible for apoptosis. DR3 activated apoptosis through triggering TRADD, FADD and caspase-8 [128], and caspase-8 further activated downstream effectors caspase-1 and caspase-3, resulting in cell death [129,130]. Similarly, cordycepin induced apoptosis in human prostate carcinoma LNCaP cells via the caspase pathway by increasing the expression of Fas, DR5, caspase-8, -9 and -3, and causing a dose-dependent increase in pro-apoptotic Bax and decrease in anti-apoptotic Bcl-2 [131]. Changes in Bax and Bcl-2 levels trigger a collapse of mitochondrial membrane potential and activation of caspase-9 and -3. A study by Balk et al. [132] revealed that cordycepin increased the levels of Fas, FasL and TRAIL (related apoptosis-inducing ligand) of U87MG cells, and decreased Bcl-2 level, indicating that cordycepin induced apoptosis via the Fas/FasL pathway. Lee et al. [25] found that *C. militaris* extract induced apoptosis by increasing the protein expression ratio of Bax/Bcl-2 and the cleavage of caspase-7, -8 and -9 in MCF-7 cells. In addition, cordycepin has been demonstrated to inhibit the proliferation of B16-BL6 mouse melanoma cells through combination with adenosine A_3_ receptor (A_3_R) on the B16 cell membrane, reducing expression of cyclin D1 protein and activating glycogen synthase kinase-3β (GSK-3β) [28]. In addition, cordycepin is also thought to induce apoptosis through A_3_R and A_2_AR. The apoptosis induction of cordycepin is possibly mediated by A_3_R in human bladder cancer T24 cells since both overexpression of A_3_R and cordycepin treatment decreased cell survival since the apoptosis-inducing effect of cordycepin is abolished with the depletion of adenosine receptors [133]. Moreover, adenosine was found to induce apoptosis and upregulate mRNAs of TNF, FADD, TRADD, and TRAIL-2 by activating caspase-3, -8 and -9 in human hepatoma HepG2 cells [8]. Ma et al. [6] found a novel apoptosis mechanism that extracellular adenosine could trigger apoptosis by increasing reactive oxygen species (ROS) production and mitochondrial membrane dysfunction in BEL-7404 liver cancer cells. Choi et al. [19] found that cordycepin induced MDA-MB-231 cells’ apoptosis through increasing translocation of Bax in the mitochondrion and triggering cytosolic release of cytochrome c and activation of caspases-9 and -3. These studies indicate that Chinese Cordyceps induces apoptosis via both the mitochondrion-mediated intrinsic pathway and extrinsic Fas/FasL and ARs pathways.

Autophagy, mediated by an intracellular suicide program, plays important roles in antitumor responses. A study by Qi et al. [51] showed that a polysaccharide named CSP from *O. sinensis* mycelia inhibited the proliferation of HCT116 human colon cancer cells through inducing apoptosis and autophagy. On one hand, CSP induced apoptosis by activating caspase-8 and -3, on the other hand, it inhibited lysosome formation, blocked autophagy flux and accumulated autophagosomes, resulting in autophagy. The further investigation revealed that signaling pathways of PI3K-AKT-mTOR and AMPK-mTOR-ULK1 were all involved. Cordycepin was also found to induce autophagic cell death and formation of a large membranous vacuole in MCF-7 human breast cancer cells, accompanied with the increase in autophagosome marker LC3-II levels [19]. Generally, autophagy occurs before apoptosis under certain stress stimuli, and autophagy will be inactivated when stress exceeds the intensity threshold or critical duration, followed by the activation of apoptosis [134].

#### 2.2.2. Blocking Cell Cycle

The cell cycle can be mainly divided into four phases, of which G1, S and G2/M phases are crucial checkpoints in the cell cycle processes. Cell cycle arrest in the G1, S and G2/M phases can lead to the inhibition of tumor cells’ proliferation and induction of apoptosis. Adenosine, cordycepin and polysaccharides were found to cause cell cycle arrest at certain checkpoints. Cordycepin inhibited the growth of 5637 and T-24 bladder cancer cells and HCT116 colon cancer cells, through G2/M cell-cycle arrest. The expression of p21WAF1 (a universal key inhibitor in regulating cell-cycle progression) was upregulated and cyclin B1, Cdc25c and Cdc2 (G2/M cell-cycle regulatory proteins) were downregulated, through the JNK1 signal pathway [50,135]. Cells blocked in the G2/M phase failed to enter mitosis, resulting in cell growth inhibition. Moreover, cordycepin induced an increase in subG1 cell number and the decrease in G1 and G2/M cell numbers and cell viability through inducing caspase-9, -3 and -7 expression as subG1 phase accumulation could be partly suppressed using a caspase inhibitor, which indicated that the caspase-9 pathway is involved in cordycepin-induced subG1 phase arrest [20].

In addition, cordycepin is a transcription and polyadenylation inhibitor and affects RNA synthesis. A study showed that cordycepin caused accumulation of the corresponding triphosphate derivative, 3′dATP, which might lead to the incorporation of analogue into nascent nucleic acid oligonucleotides and RNA synthesis inhibition [18,136]. This might illustrate that cordycepin affects the cell cycle from another perspective. An extract of *Cordyceps cicadae* was identified as a nucleoside mixture containing adenine, adenosine, uridine and N6-(2-Hydroxyethyl)-adenosine, induced S phase arrest in human gastric cancer SGC-7901 cells, which was related to downregulation of CDK2 expression and upregulation of expression of transcription factor E2F1 (cyclin/CDK complexes, regulating G1/S phase transition), cyclin A2 and cyclin E [137].

#### 2.2.3. Inhibiting Migration, Invasion and Metastasis

Metastasis refers to the movement of cancer cells from primary tumor sites to other organs and tissues and is the end result of multiple interactions including invasion between the tumor and host, indicating the uncontrolled spread of the tumor cells. Epithelial–mesenchymal transition (EMT)-related proteins such as matrix metalloproteinases (MMPs) play an important role in metastasis. For example, MMP-2 and MMP-9 can lead to the degradation of extracellular matrix (ECM) components and tissue invasion [138,139,140]. A study showed that cordycepin inhibited 5637 and T-24 cells’ invasion through decreasing MMP-9 expression and the transcriptional activity of activator protein-1 (AP-1), which were identified by gel-shift assay as cis-elements for TNF-α activation of the MMP-9 promoter via the NF-κB/MMP-9 pathway [94]. In addition, a novel polysaccharide CME-1 isolated from *O. sinensis* was found to inhibit migration of B16-F10 melanoma cells, and the mechanism was that CME-1 reduced MMP-1 expression and downregulated the phosphorylation level of ERK1/2 and p38 MAPK [49]. 

Angiogenesis is vital for organ growth and repair, and essential for tumor growth. The vascular endothelial growth factor (VEGF) family plays an important role in angiogenesis. VEGF, a key angiogenic growth factor, has a higher expression level in tumor tissues and can accelerate the differentiation, proliferation, and migration of endothelial cells. Chinese Cordyceps has been demonstrated to inhibit the VEGF/VEGFR2 signaling pathway and exert antiangiogenesis function [116]. Moreover, the overexpression of proto-oncogenes c-Myc and c-Fos may promote tumor cell proliferation under growth-promoting stimulation. c-Myc, encoding a ubiquitous transcription factor and promoting cell division, is related to apoptosis and the occurrence and development of various tumors. c-Fos, essential for cell proliferation, can upregulate the cell cycle by induction of cyclin D1 [141]. c-Fos is expressed at a low level in normal cells while it is overexpressed in tumor cells. Yang et al. [120] found that EPSF isolated from C.sinensis could downregulate the expression of VEGF, c-Myc and c-Fos, which was the important factor to inhibit tumor growth, invasion and metastasis.

## 3. Discussion

In recent years, the traditional therapy for cancer has become an attention direction of researchers, and many researchers believe that traditional therapy is a potential new therapy. The pathogenesis of cancer is diverse and complex, and Chinese Cordyceps has many active ingredients and diverse extracts, which can inhibit the growth of various tumors and prevent or overcome metastasis through various pathways (Figure 3). It is well known that improving self-immunity can lay a good foundation for fighting and treating many diseases. Chinese Cordyceps has a long history of use in China, and much evidence suggests that Chinese Cordyceps, acting as an immune response activator, is used for the treatment of a variety of diseases including cancer. Increasing studies have shown that Chinese Cordyceps has immunomodulatory, anti-inflammatory and antioxidant activities that affect the immune system and TME in various ways. The polarization and remodeling of the phenotype of immune cells (such as T cells and macrophages) by Chinese Cordyceps have effects on cytokine production in TME, which may affect tumor progression. The anticancer ability of Chinese Cordyceps has been the subject of research for nearly 60 years and its antitumor effect has been confirmed in cancer cells or mouse cancer models alone or in combination with other drugs. The research on clinical application still needs more attention. 

## 4. Conclusions

In conclusion, Chinese Cordyceps has significant antitumor activity and immunomodulatory activity. On the one hand, it can directly act on tumor cells to kill tumor cells or inhibit tumor growth and effectively attenuate tumor cell metastasis. On the other hand, Chinese Cordyceps can change the tumor microenvironment and enhance antitumor immune responses by downregulating the expression of immunosuppressive factors and upregulating the expression of pro-inflammatory factors, thereby improving the antitumor function. These findings may provide therapeutic strategies for treating cancer.

## Figures and Tables

**Figure 1 molecules-27-06576-f001:**
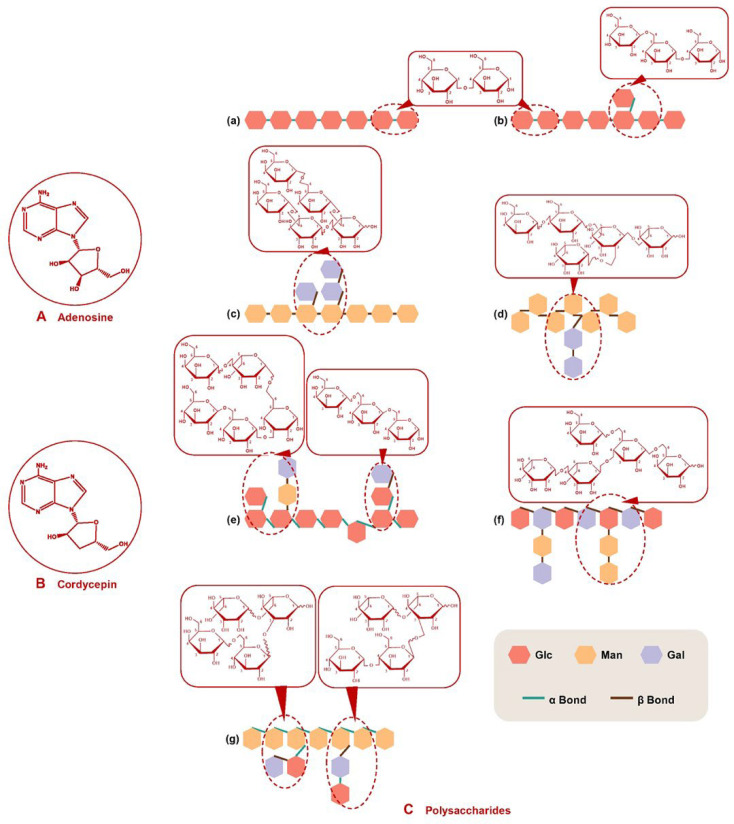
Chemical structure of adenosine (**A**), cordycepin (**B**) and polysaccharides (**C**) of *Cordyceps* spp. Illustration of the chemical structure of several *Cordyceps* polysaccharides: (**a**) (α1→4)-glucan; (**b**) (α1→6)-branched, (α1→4)-glucan; (**c**) (β1→6)-branched, (β1→4)-galactomannan; (**d**) (β1→4)-(β1→6)-branched, (β1→2)-(β1→6)-galactomannan; (**e**) (β1→4)-(β1→6)-(α1→6)-branched, (α1→3)-galactoglucmannan; (**f**) (β1→4)-branched, (β1→6)-galactoglucmannan; (**g**) (α1→4)-(β1→6)-branched, (α1→6)-galactoglucmannan.

**Figure 2 molecules-27-06576-f002:**
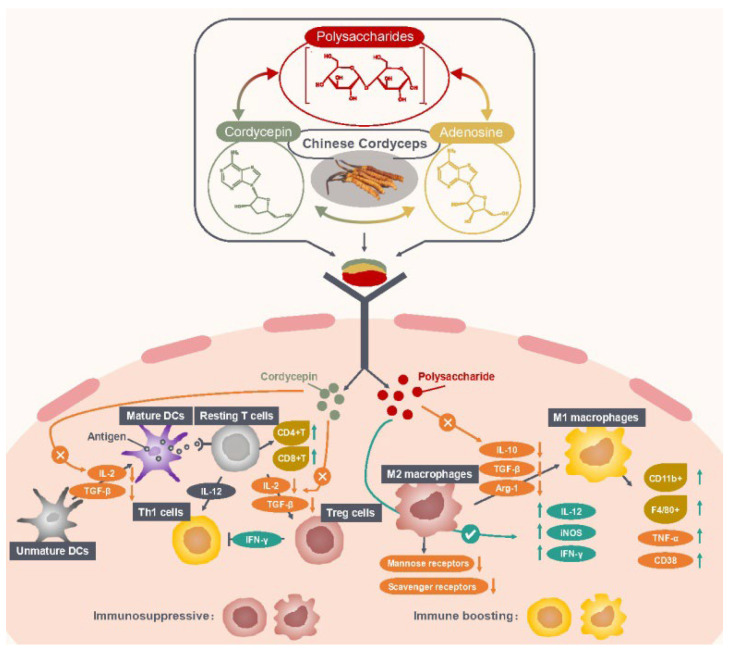
Mechanism of Chinese Cordyceps regulating immune cells in TIM. Abbreviations: Arg-1, arginase-1; CD38, CD11b^+^ and F4/80^+^, M1 macrophage markers; IFN-γ, interferon-gamma; IL, interleukin; iNOS, inducible nitric oxide synthase; TGF-β, transforming growth factor-beta; TIM, tumor immune microenvironment; TNF-α, tumor necrosis factor-alpha.

**Figure 3 molecules-27-06576-f003:**
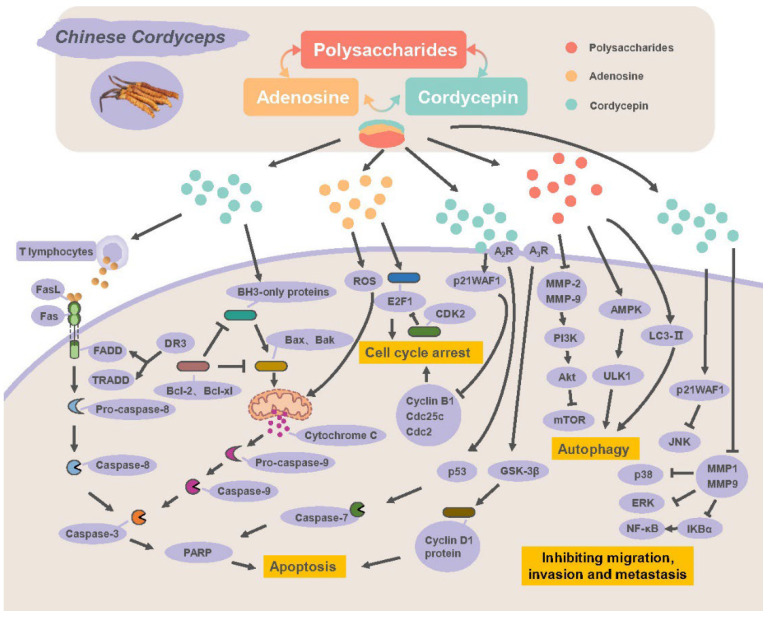
Possible mechanisms of antitumor activity of Chinese Cordyceps.

**Table 1 molecules-27-06576-t001:** Bioactive components and bioactivities of Chinese Cordyceps.

No.	Bioactive Components	Pharmacological Effects	Ref.
1	Adenosine	Antitumor activity	[5,6,7,8]
		Attenuation of chronic heart failure	[9]
		Anti-inflammation	[10,11,12,13,14]
		Immunomodulatory activity	[12,15]
2	Inosine	Anti-inflammation	[16]
3	Guanosine	Seizure prevention	[17]
		Immunomodulatory activity	[15]
4	Cordycepin	Antitumor activity	[18,19,20,21,22,23,24,25,26,27,28,29]
		Antibacterial activity	[20]
		Treatment for ischemic/reperfusion (IR) injury	[21]
		Anti-inflammation	[22,23,24,25,26,27,28,29,30,31,32,33,34,35,36,37]
		Immunomodulatory activity	[38,39,40]
		Antioxidant activity	[41,42]
		Cholesterol lowering effect	[43]
		Anti-fibroblast activity	[44]
5	Cordycepic acid	Diuretic effect	[45]
		Attenuating postreperfusion syndrome	[46]
		Anti-fibrosis and anti-inflammation	[47]
6	Polysaccharides	Antitumor activity	[48,49,50,51,52]
		Immunomodulatory activity	[48,53,54,55,56,57,58,59,60,61]
		Anti-inflammation	[56,62,63]
		Antioxidant activity	[61,64,65]
		Antiviral activity	[66]
		Protective effects on kidney	[67,68]
		Hypoglycemic effect	[69]
7	Cordymin	Analgesic effect	[70]
		Anti-inflammation	[71]
		Antioxidant	[71]
		Hypoglycemic effect	[72]
8	Cordycedipeptide A	Antitumor activity	[73]
9	Tryptophan	Sedative hypnotic effect	[74]
10	Fibrinolytic enzymes	Treatment for thrombosis	[75,76]
11	Ergosterol	Cytotoxicity	[77]
		Anti-inflammation	[78]
		Anti-fibroblast activity	[79]
		Antiviral activity	[80]
12	β-Sitosterol	Cytotoxicity	[77,81]
13	5*α*,8*α*-epidioxy-22*E*-ergosta-6,22-dien-3*β*-ol	Cytotoxicity	[77]
14	5*α*,8*α*-epidioxy-22*E*-ergosta-6,9(11),22-trien-3*β*-ol	Cytotoxicity	[77]
15	5*α*,6*α*-epoxy-5*α*-ergosta-7,22-dien-3*β*-ol	Cytotoxicity	[77]
16	H1-A	Cytotoxicity	[82]
17	Cordysinin A	Anti-inflammatory	[83]
18	Cordysinin B	Anti-inflammatory	[83]
19	Cordysinin C	Anti-inflammatory	[83]
20	Cordysinin D	Anti-inflammatory	[83]
21	Cordysinin E	Anti-inflammatory	[83]

**Table 2 molecules-27-06576-t002:** Polysaccharides that originated from *Cordyceps* spp.: chemical structure and immunomodulatory and antitumor activities.

No.	Name	MW	Components	Glycosyl Linkage and Branches (Characteristic Signals)	Bioactivities	Source	Ref.
1	AEPS-1	36 kDa	Glc*p*:GlcU*p* = 8:1 (M ratio), plus a trace amount of mannose	A main chain of (1→3)-linked α-d-Glc*p* with α-d-Glc*p* and α-d-GlcU*p* branches attached to the main chain by (1→6) glycosidic bonds at every seventh α-d-Glc*p* unit	Anti-inflammatory; immunomodulatory	Mycelial fermentation of *C. sinensis* (Cs-HK1)	[93]
2	EPS	104 kDa	Man:Glc:Gal = 23:1:2.6 (M ratio)		Immunomodulatory	Mycelial fermentation of *C. sinensis* (G1)	[58]
3	NCSP-50	976 kDa	Glucose	A main chain of (1→4)-linked α-d-Glc*p* with α-d-Glc*p* branch attached to the C-6	Immunomodulatory	*C. sinensis*	[59]
4	CSP	28 kDa	Gal:Glc:Man:Ara:GalA = 36.40:28.99:24.81:3.34:7.55 (percentage ratio)	A main chain of (1→4)-linked α-d-Glc and (1→4)-linked α-d-Gal	Antitumor	Cultured mycelia of *C. sinensis*	[51,65]
5	CME-1	27.6 kDa	Man:Gal:Glc = 39.1:59.2:1.7 (M ratio)	A backbone of (1→4)-linked β-d-Man with Gal branches attached to the O- 6	Antitumor	Cultured mycelia of *C. sinensis*	[49]
6	APSP		Man:Glc:Gal = 3.5:1:1.5 (M ratio)		Immunomodulatory	Mycelia of liquid cultured *C. sinensis*	[53]
7	PLCM (CPSN Fr II)	36 kDa	Man:Gal:Glc:Protein:Hexosamine:Uronic acid = 65.12:28.72:6.12:0.20:0.06:0.29 (percentage ratio)	A backbone of (1→2)-, (1→6)-linked β-d-Man with (1→4)-linked β-d-Gal branches attached to the O- 6	Immunomodulatory	*C. militaris* liquid culture broth	[55,94]
8	CMP-III	4.796 × 10^4^ kDa	Glc:Man:Gal = 8.09:1.00:0.25 (M ratio)	A backbone of (1→4)-linked α-d-Glc with (1→4,6)-linked α-d-Man and (1→2,6)-α-d-Man branches attached to the O- 6	Immunomodulatory	Cultured fruiting bodies of *C. militaris*	[54]
9	CMPB90-1	5.8 kD	Gal:Glc:Man = 3.04:1:1.45 (M ratio)	A main chain of (1→6)-linked α-d-Glc and (1→3)- linked α-d-Glc, with branching at O-6, which consists of (1→4)-linked β-d-Man and (1→6)-linked α-d-Glc, respectively, and β-d-Man as the terminal unit	Immunomodulatory	Cultured fruiting bodies of *C. militaris*	[95]
10	CPMN Fr III	210 kDa	Glc:Gal:Man = 9.17:18.61:72.22 (M ratio)	A backbone of (1→6)- linked β-d-Man and (1→6)- linked β-d-Glc with branches of (1→4)- linked β-d-Man terminated with d-Gal and d-Man, respectively	Immunomodulatory	Cultured mycelia of *C. militaris*	[50]
11	HS002-II	44 kDa	D-Man:D-Rib:L-Rha:D-GlcUA:D-GalUA:D-Glc:D-Gal:D-Xyl:L-Ara = 6.47:2.27:1.25:0.69:0.42:65.89:16.17:2.13:4.26 (M ratio) polysaccharide:protein = 57.9:42.1 (percentage ratio)	A long backbone of (1→3)-linked α-d-Rib*f*, (1→4)-linked α-d-Xy*lp* and approximately 1/31 of (1→4)-linked β-d-Glc*p*, which was substituted at C-6. The two branches were (1→6)-linked β-d-Man*p* and (1→6)-linked β-d-Gal*p* terminated with α-L-Ara*p*, respectively	Immunomodulatory	Mycelial fermentation of *Hirsutella sinensis* Liu, Guo, Yu and Zeng	[96]
12	P70-1	42 k Da	Man:Gal:Glc = 3.12:1.45:1.00 (M ratio)	A backbone of (1→6)-linked α-d-Man*p* with branching points at O-3, and the branches composed of (1→4)-linked α-d-Glc*p* and (1→6)-linked β-d-Gal*p*, and terminated with β-d-Gal*p* and α-d-Glc*p*	Antioxidant	Fruiting bodies of cultured *C. militaris*	[97]
13	CPS-1	23 kDa	Rha:Xyl:Man:Glc:Gla = 1:6.43:25.6:16.0:13.8 (M ratio)	Composed of (1→2)-linked Man, (1→4)-linked Xyl and (1→2)-linked or (1→3)-linked Rha or Gal	Anti-inflammatory	Cultured *C. militaris*	[63]
14	AIPS	1.15 × 10^3^ kDa	Glucose	α-d-(1→4) glucan	Antitumor	Mycelial fermentation of *C. sinensis* (Cs-HK1)	[98]

Ara, arabinose; Ara*p*, arabinopyranosyl; Gal, galactose; GalA, galacturonic acid; GalUA, galacturonic acid; Glc, glucose; Glc*p*, glucopyranose; GlcUA, glucuronic acid; GlcU*p*, pyrano-glucuronic acid; Man, mannose; Man*p*, mannopyranosyl; MW, molecular weight; Rha, rhamnose; Rib, ribose; Rib*f*, ribofuranosyl; Xyl, Xylose; Xyl*p*, xylopyranosyl.

**Table 3 molecules-27-06576-t003:** Antitumor immunity effects and mechanisms of Chinese Cordyceps in various models.

Bioactive Component	Pharmacological Effects	Models	Major Mediating Signaling Pathways	Mechanism of Action	Ref.
Cordycepin	↑Antitumor immunity responses ↓CT 26 cell migration ↑CT 26 cell apoptosis	CT 26 cells in mice		↑CD4^+^ T, CD8^+^ T cells ↑NK cells ↑M1 macrophages ↑CD11b^+^, F4/80^+^ ↓CD47	[38]
JLM 0636 (cordycepin-enriched extract of *C. militaris*)	↑Th 1 cells ↑Immune responses ↓Treg cells ↓Immunosuppression	FM3A murine breast cancer cells, derived from C3H/He mouse		↑CD8^+^ T cells ↑IFN-γ ↓CD4^+^CD25^+^ T cells ↓IL-2 ↓TGF-β	[39]
WECS (Nucleoside extract of *C. sinensis*)	↓MDA-MB-231 cells ↓4T1 cells ↑M1 macrophages ↑Immune responses	MDA-MB-231, 4T1 breast cancer cells co-cultured with macrophages	NF-κB	↑CD38 ↑iNOS ↑IL-1β ↑IL-12p70 ↑TNF-α ↑IL-6 ↑IFN-γ ↑NO	[102]
EPSP	↑M1 macrophages ↑Spleen lymphocyte ↑Immune response ↓Tumor migration	B16 melanoma-bearing mice		↓Bcl-2	[52]
APSF	↑M1 macrophages ↑Immune response ↓M2 macrophages ↓Immunosuppression	Ana-1 mouse macrophages co-cultured with H22 cells	NF-κB	↑TNF-α ↑IL-12 ↑iNOS ↓IL-10 ↓SR ↓MR	[103]
CMPB90-1	↑M1 macrophages ↑Immune response ↓M2 macrophages ↓Immunosuppression	IL-4, tumor cell supernatant-induced RAW264.7 cells	NF-κB Akt MAPK (p38 and ERK)	↓IL-10 ↓TGF-β ↓Arg-1 ↑IL-12 ↑iNOS	[48]

Akt, serine/threonine kinase; Arg-1, arginase-1; ERK, extracellular-signal-regulated kinases; IFN-γ, interferon-gamma; IL, interleukin; iNOS, inducible nitric oxide synthase; IPS, intracellular polysaccharide; MAPKs, mitogen-activated protein kinases; MR, mannose receptor; NK, natural killer cell; NF-κB, nuclear factor kappaB; NO, nitric oxide; TGF-β, transforming growth factor-beta; TNF-α, tumor necrosis factor-alpha; SR, scavenger receptor.

**Table 4 molecules-27-06576-t004:** Antitumor effects and mechanisms of Chinese Cordyceps in various cancer models.

Cancer	Bioactive Component	Pharmacological Effects	Cell line	Major Mediating Signaling Pathways	Mechanism of Action	Ref.
Bladder cancer						
	Cordycepin	↓Migration and invasion	TNF-α-induced 5637 and T-24 cells	NF-κB AP-1	↓MMP-9	[23]
Breast cancer						
	Cordycepin	↑Apoptosis	MDA-MB-231 cells	Caspase	↑Bax (mitochondria) ↑Cytochrome c (cytosol) ↑PARP ↑c-caspases-9, -3 ↑DNA fragmentation	[19]
	Cordycepin	↑Autophagy	MCF-7 cells	Autophagy	↑LC3-II ↑Autophagosome-like structure	[19]
	Cordycepin	↑Apoptosis	MDA-MB-435 and T47D cells		↑DNA fragmentation ↑Histone γH2AX ↓RNA synthesis	[26]
	*C. militaris* extract	↑Apoptosis	MCF-7 cells	Caspase	↑Bax/Bcl-2 ↑c-caspase-7, -8	[25]
	Cordycepin	↑Apoptosis	C6 glioma cells	A_2_AR Caspase	↑Caspase-7 ↑p-p53 ↑PARP	[110]
Cervical cancer						
	Cordycepin	↑Apoptosis ↓Cell cycle	SiHa cells HeLa cells		↓CDK-2 ↓Cyclin-E1 ↓Cyclin-A2 ↑ROS	[111]
	CCP (*C. cicadae* polysaccharides)	↑Apoptosis ↓Cell cycle	hela cells	Akt	↑Bak ↑Bax ↑Caspase-3, -7, -9 ↓P21 ↓P27 ↓CDK2 ↓Cyclin E1 ↓Cyclin A2 ↓Bcl-2 ↓Bcl-xl ↓PARP	[112]
Colon cancer						
	CSP	↑Autophagy, ↑Apoptosis	HCT116 cells	Autophagy mTOR Caspase	↑LC3B-II ↑Caspase-8, -3	[51,65]
	Cordycepin	↓Cell cycle	HCT116 cells	JNK MAPK	↑p21WAF1 ↓Cyclin B1 ↓Cdc25c ↓Cdc2	[24]
Colorectal cancer						
	*C. militaris* extract	↓Cell cycle	RKO cells		↑Bax ↑Bim ↑Bak ↑Bad ↑PARP ↑p-p53 ↑c-caspase -9, -3	[113]
Gastric cancer						
	Cordycepin	↑Apoptosis	AGS cells	PI3K/Akt	↑Caspase-9, -3, -7 ↑Bax ↓Bcl-2	[114]
	CECJ (*C. jiangxiensis* extract)	↑Apoptosis ↓Cell cycle	SGC-7901 cells	Caspase	↑Caspase-3	[115]
Liver cancer						
	Adenosine	↑Apoptosis	HepG2 cells	Caspase	↑TNF ↑TRADD ↑TRAIL-R2 ↑FADD ↑Caspase-9, -8, -3	[8]
	Adenosine	↑Apoptosis	BEL-7404 cells	Caspase	↑Caspase-8, -9, -3 ↑c-PARP ↑Bak ↑Mcl1 ↑Bcl-xl	[6]
	Adenosine	↑Apoptosis	HuH-7 Fas-deficient cells	Caspase	↑AMP ↓Caspase-3, -8 ↓c-FLIP	[7]
	CMF (*C. militaris* extract)	↓Migration and invasion ↓Tumor growth	SMMC-7721 cells	Akt ERK	↓p-VEGFR2 ↓p-Akt ↓p-ERK	[116]
Lung cancer						
	AECS1, AECS2 (*C. sinensis* nucleosides extract)	↓Tumor growth	Lewis xenograft mouse	Akt NF-κB	↓p-Akt ↓MMP2 ↓MMP9 ↓p-IκBα ↓TNF-α ↓COX-2 ↓Bcl2 ↓Bcl-xl ↑Bax	[117]
	CS (*C. sinensis* extract)	↑Apoptosis	H157 NSCLC cells		↓VEGF ↓bFGF	[118]
	*C. militaris* extract	↑Apoptosis ↓Cell cycle	NCI-H460 cells		↑P53 ↑P21 ↑53BP1	[119]
Mouse melanoma						
	Cordycepin	↓Proliferation	B16-BL6 cells	A_3_R	↑GSK-3β ↓Cyclin D1 protein	[28]
	CME-1	↓Tumor migration	B16-F10 cells	NF-κB MAPK (ERK and p38)	↓MMP-1	[49]
	EPSP	↓Tumor migration	B16 cells		↓c-Myc ↓c-Fos ↓VEGF	[120]
Myeloma cancer						
	Cordycepin	↑Apoptosis	MM.1S cells	Caspase	↑Caspase-9, -3, -8 ↓RNA synthesis	[8]
Oral cancer						
	CMP (*C. militaris* polysaccharides)	↑Apoptosis ↓Cell cycle	4NAOC-1 cells	STAT3 ERK	↓ki-67 ↓EGFR ↓IL-17A ↓Cyclin B1 ↓DNA synthesis	[121]
	WECM (*C. militaris* extract)	Apoptosis ↓Cell cycle	SCC-4 cells		↓PCNA ↓VEGF ↓Caspase-3 ↓c-fos	[122]
Ovarian cancer						
	CME (*C. militaris* extract)	↑Apoptosis ↓Migration	SKOV-3 cells	NF-κB	↓TNF-1R ↓Bcl-2 ↑Bcl-xl	[123]
		↑Autophagy ↓Tumor growth	A2780 and OVCAR3 cells	ENT1-AMPK-mTOR	↑LC3II/LC3I ↑p-AMPK	[124]
Prostate cancer						
	Cordycepin	↓Migration and invasion	LNCaP cells	PI3K/Akt	↑TIMP-1 ↑TIMP-2 ↓MMP-2 ↓MMP-9	[21]
Testicular cancer						
	Cordycepin	↑Apoptosis ↓Cell cycle	MA-10 cells	Caspase	↑Caspase-9, -3, -7 ↑DNA fragmentation	[20]

AMP, activated protein kinase; A_2_AR, adenine 2A receptor; A_3_R, adenine 3 receptor; c-FLIP, cellular FADD-like interleukin-1β-converting enzyme inhibitory protein; c-PARP, cleaved-poly ADP-ribose polymerase; COX-2, cyclooxygenase-2; c-Fos, c-Myc, cellular proto-oncogenes; cyclin B1, Cdc25c and Cdc2, cell cycle regulatory proteins; ERK, extracellular signal-regulated kinases; FADD, fas-associated death domain; JNK, Jun N terminal kinase; LC3-II, the lipidated form of LC3B; MMP, mitochondrial membrane potential; mTOR, mechanistic target of rapamycin; MAPK, mitogen-activated protein kinases; p-Akt, phosphorylated serine/threonine kinase; p21WAF1, cyclin-dependent kinase inhibitor; TNF, tumor necrosis factor; TRADD, TNF receptor-associated death domain; TRAIL-R2, TNF-related apoptosis-inducing ligand receptor 2; VEGF, vascular endothelial growth factor.

## Data Availability

Not applicable.

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
