# Peer review of "Chinese Cordyceps: Bioactive Components, Antitumor Effects and Underlying Mechanism—A Review"

_molecules, 2022, doi:10.3390/molecules27196576_

Round 1

Reviewer 1 Report

This work is interesting for accumulating the benefit of traditional plants or herbs of China for health functionality. All section was well written. However, some points of this work need more clarification by the author and/or added more detail before accepting.

1. The abstract graphic and Figure 1 needs to increase the resolution

2. All tables, the narrow for up or down should be modified. It seems not to match the detail.

3. Author may revise the sentence in the manuscript that is almost similar to the previous report following the Plagiarism checking file attachment.

4. For the conclusion section, too many details were put herewith. The authors need to summarize the vital point of the work only. The minor point may be moved to the sub-topic in each part for discussion or related to that section.

For the part of Antitumor activities and their participation in molecular mechanisms (Line 130), how about the MCP-1 work/interact with Chinese Cordyceps? These cytokines may affect the immune pathway and can be stimulated or suppressed the other immune cytokines' activity. If possible, may author should discuss this point for a while.

Author Response

Response to Reviewer 1 Comments

 1. Reviewer: The abstract graphic and Figure 1 needs to increase the resolution.

Response: I agree with reviewer’s suggestion, and have revised the abstract graphic and Figure 1.

  1. Reviewer: All tables, the narrow for up or down should be modified. It seems not to match the detail.

Response: All arrows have been resized and embedded in Table 3 and Table 4, one-to-one correspondence.

  1. Reviewer: Author may revise the sentence in the manuscript that is almost similar to the previous report following the Plagiarism checking file attachment.

Response: I agree with reviewer’s suggestion, and have revised all the sentences which is presented as follows:

Page 1 line 8, “Chinese Cordyceps is a well-known medical mushroom with multiple biological effects” is changed to “Chinese Cordyceps is a valuable source of natural products with various therapeutic effects”.

Page 2 line 32, “Chinese” is changed to “China”.

Page 2 line 33, “is the entomopathogenic fungus coming out of the mummified body of a caterpillar” is changed to “is composed of the caterpillar and the entomopathogenic fungus”.

Page 2 line 37, “In 1843, the British Mycologist named Berkely first described this fungus as Sphaeria sinensis Berk. Later in 1878, the Italian scholar named Saccardo named Cordyceps derived from China officially as Cordyceps sinensis” is changed to “This fungus was first named as Sphaeria sinensis Berk by the British Mycologist Berkely in 1843, and later Cordyceps derived from China was officially recognized as Cordyceps sinensis”.

Page 2 line 40, “are all using” is changed to “use”.

Page 2 line 43, “traditionally Chinese Cordyceps is considered a parasitic complex of a fungus (Hirsutella sinensis or O. sinensis) and a caterpillar” is deleted.

Page 2 line 45, “is described as” is changed to “refers to” and “flask” is deleted and in line 46, “Prenemycetes” is changed to “Pyrenemycetes”.

Page 2 line 50, “Cordyceps sinensis, Cordyceps militaris, Cordyceps sobolifera, Cordyceps cicadicola, C. subsessilis, Cordyceps liangshanesis, and Cordyceps ophioglossoides” is changed to “C. sinensis (=O. sinensis), C. militaris, C. sobolifera, C. cicadicola, C. subsessilis, C. liangshanesis and C. ophioglossoides”.

Page 2 line 53, “the” is changed to “two”.

Page 2 line 56, “mostly distributed in Tibet, Qinghai, Yunnan, Sichuan, Gansu and other provinces” is changed to “mainly distributed in Tibet, Qinghai, Yunnan, Sichuan and Gansu provinces”.

Page 2 line 62, “various pharmaceutical properties” is changed to “multiple pharmaceutical functions”, and “anti-inflammatory, anti-tumor” is changed to “antiinflammatory, antitumor”, and in line 63, “activities” is deleted.

Page 2 line 63, “antihypertensive and antiaging activities [8,9], and are used for the treatment of respiratory and liver and lung diseases, cardiovascular diseases, hyperlipidemia, diabetes mellitus and kidney failure” is changed to “antihypertensive, antihyperlipidemia, antiaging, protection of liver and kidney and improving cardiovascular function [8-12]”.

Page 2 line 69, “In this review, firstly we compared and analyzed the characteristics of the structural and physicochemical properties of the main components, and then reiterated their antitumor effects and the mechanisms of action from the aspect of immunotherapy” is changed to “In this review, firstly we summarized the structural characteristics and physicochemical properties of the main components (adenosine, cordycepin and polysaccharides), and then evaluated recent antitumor studies on Chinese Cordyceps to determine its future perspective as an antitumor drug and elucidate its possible antitumor mechanisms of action.”

Page 2 line 74, “high consumption of foods rich in bioactive compounds with different phytochemicals is beneficial to human health.” is changed to “high intake of foods rich in bioactive compounds is beneficial to human health.”

Page 3 line 82, “(ie. B1, B2, B12, E and K)” is deleted.

Page 3 line 85, “ingredients” is changed to “components”.

Page 3 Table1.

No.13,“5a,8a-epidioxy-22E-ergosta-6,22-dien-3b-ol” is changed to “5α,8α-epidioxy-22E-ergosta-6,22-dien-3β-ol”;

No.14,“5a,8a-epidioxy-22E-ergosta-6,9(11),22-trien-3b-ol” is changed to “5α,8α-epidioxy-22E-ergosta-6,9(11),22-trien-3β-ol”;

No.15,“5a,6a-epoxy-5a-ergosta-7,22-dien-3b-ol” is changed to “5α,6α-epoxy-5α-ergosta-7,22-dien-3β-ol”.

Page 4 line 87, “Adenosine is regarded as a marker for quality control of Chinese Cordyceps in Chinese Pharmacopoeia” is changed to “Adenosine is regarded as an important marker reflecting the quality of Chinese Cordyceps in Chinese Pharmacopoeia”.

Page 4 line 92, “, with difference of only hydroxyl group, lacking in the 3′ position of its ribose moiety” is changed to “similar in structure to adenosine except for lacking a hydroxyl group in the 3′ position of its ribose moiety”. And “It was first isolated from C. militaris in 1951.” is deleted.

Page 4 line 95, “± 0.84” is deleted.

Page 4 line 96, “, but not” is changed to “while insoluble”.

Page 4 line 97, “So many researchers use sterilized saline or phosphate-buffered saline (PBS) as solvent. Several extract methods of nucleosides are used: organic solvent pressurized liquid extraction, boiling water extraction, and ambient temperature water extraction [99, 100].” is changed to “Usually Chinese Cordyceps is grinded into powder and extracted with methanol aqueous solution or distilled water by ultrasonic machine or by soxhlet extractor to get nucleosides [99, 100]. And phosphate-buffered saline (PBS) is used as the solvent.”

Page 4 line 109, “Polysaccharides are the main contributor towards the pharmacological properties” is changed to “Polysaccharides are another important biologically active component”.

Page 4 line 110, “it is challenging to take these polysaccharides as markers for quality because of its structural complexity and unstable molecular weight.” is deleted.

Page 4 line 112, “the most common class of immunomodulating and antitumor polysaccharides of medicinal fungi comprise of the glucans with various glycosidic linkages” is changed to “medicinal fungal polysaccharides with immunomodulatory and antitumor effects are the most common glucans linked by various glycosidic bonds”.

Page 5 line 115, “are the most common bioactive polysaccharides in the fruiting bodies and mycelia of C. sinensis” is changed to “are very important bioactive polysaccharides in Chinese Cordyceps”.

Page 5 line 117, “molecular weight. It has been reported that polyglucans with higher MW (10-1000 kDa) have greater water solubility” is changed to “MW. The larger the MW (10-1000 kDa) of the polysaccharides is, the greater the water solubility and the better the biological activity.

Page 5 line 119, “Water extraction combined with alcohol precipitation is the most common and effective method to extract active polysaccharides. Chinese Cordyceps polysaccharides include extracellular polysaccharide (EPS) and intracellular polysaccharide (IPS). EPSs resource from the fermentation broth of submerged Chinses Cordyceps fungi, while IPSs resource from fruiting bodies of Chinses Cordyceps and cultured mycelium.” is changed to “Polysaccharides have good water solubility, and water extraction combined with alcohol precipitation is a very effective method to extract active polysaccharides. Cordyceps polysaccharides include two forms: extracellular polysaccharide (EPS), mainly resourcing from the fermentation broth of submerged Chinses Cordyceps fungi, and intracellular polysaccharide (IPS), mainly resourcing from fruiting bodies of Chinses Cordyceps and cultured mycelium.”

Page 5 line 123, “The structure feature and bioactivities of polysaccharides of Chinese Cordyceps are shown in Table 2” is changed to “The chemical structure and immunomodulatory and antitumor activities of Polysaccharides originated from Cordyceps spp. are shown in Table 2.”

Line 125 Table 2,

No.1 “a linear backbone of (1→3)-linked α-D-Glcp residues with two branches, α-D-Glcp and α-D-GlcUp, attached to the main chain by (1→6) glycosidic bonds at every seventh α-D-Glcp unit” is changed to” a main chain of (1→3)-linked α-D-Glcp with α-D-Glcp and α-D-GlcUp branches attached to the main chain by (1→6) glycosidic bonds at every seventh α-D-Glcp unit.”

No.2 “a main chain of (1→4)-linked α-D-Glcp with a single α-D-Glcp branch substituted at C-6.” is changed to “a main chain of (1→4)-linked α-D-Glcp with α-D-Glcp branch attached to the C-6.”

No.6 “Man: Glc: Gal = 3.5:1:1.5” is changed to “Man: Glc: Gal = 3.5: 1: 1.5”.

No.9 “a main chain of (1→6)-α-D-Glc and (1→3)-α-D-Glc, with branching at O-6, which consists of  (1→4)-linked β-D-Man and (1→6)-linked α-D-Glc, respectively, and β-D-Man as the terminal unit.” is changed to “a main chain of (1→6)-linked α-D-Glc and (1→3)-linked α-D-Glc, with branching at O-6, which consists of  (1→4)-linked β-D-Man and (1→6)-linked α-D-Glc, respectively, and β-D-Man as the terminal unit.

No.14 “1150 kDa” is changed to “1.15 ×103 kDa”.

Line 134, “Chinese Cordyceps shows a biphasic regulation on immune cells phenotype to enhance antitumor immune responses in tumor immune microenvironment” is changed to “Chinese Cordyceps has a biphasic regulatory effect on immune cell phenotype, and can increase antitumor immune activity in tumor immune microenvironment”.

Line 137, Table 3

[48],  “CD4+ T, CD8+T, NK Cells, and M1 macrophages“ is changed to “ CD4+ T, CD8+T, NK and M1 macrophages“.

[49],  “ CD8+ T cells“ is changed to “ CD8+ T cells“

[112], “ breast cancer cells MDA-MB-231 or 4T1 cells co-cultured with macrophages cells” is changed to “MDA-MB-231, 4T1 breast cancer cells co-cultured with macrophages”.

[58], “IL-4 induced RAW264.7 cells; tumor cells supernatant (TCS)-induced RAW264.7 cells” is changed to “IL-4, tumor cells supernatan induced-RAW264.7 cells”.

Line 143, “inhibit the differentiation T cells into regulatory T cells (Tregs, a suppressive phenotype of T cells)” is changed to “inhibit the differentiation of T cells into regulatory T cells (Treg, a suppressive phenotype of T cells)”

Line 146, “secretion” is changed to “secretions”, and “was” is changed to “were”

Line 150, “into“ is changed to “at“.

Line 151, “depending on the stimulus signals transferred from tumor microenvironment (TME)“ is changed to “ based on the stimulatory signals from tumor microenvironment (TME)“

Line 153, “ M2 macrophages have been regarded as a potential target for cancer therapies“ is changed to “currently M2 macrophages are potential targets for the treatment of cancer”.

Line 153, “ Activated M2 macrophages release immunosuppressive cytokine (ie, IL-4 and IL-10) and promote tumor progression by recruiting Th2 and Treg cells and suppressing the immune system” is changed to “Activated M2 macrophages would suppress the immune system and promote tumor progression by releasing immunosuppressive cytokines (i.e. IL-4 and IL-10) and recruiting Th2 and Treg cells”.

Line 159, “ A research by Chen et al. [113] showed that a polysaccharide APSF isolated from the fruiting bodies of O. sinensis reversed M2“ is changed to “ A study by Chen et al. [113] showed that APSF, a polysaccharide isolated from the fruiting bodies of O. sinensis, reversed M2”.

Line 171, “Chinese Cordyceps and its bioactive constituents can exert antitumor immunity effects by upregulating immune responses” is changed to “These findings demonstrated that Chinese Cordyceps and its bioactive constituents could promote immune cells antitumor function by enhancing immune responses”.

Line 175, “A graphic summary of antitumor immunity of immune cells regulated by Chinese Cordyceps” is changed to “Mechanism of Chinese Cordyceps on regulating immune cells in TIM. Abbreviations: Arg-1, arginase-1; CD38, CD11b+ and F4/80+, M1 macrophages markers; IFN-γ, Interferon-gamma; IL, interleukin; iNOS, inducible nitric oxide synthase; TGF-β, transforming growth factor-beta; tumor immune microenvironment; TNF-α, tumor necrosis factor-alpha.”.

Line 179, “Programmed cell death, involving apoptosis and autophagy, is” is changed to “Apoptosis is one of the programmed cell death and ”.

Line 182, “Fas/FasL is an important signal pathway in the molecular regulation of apoptosis.” is changed to “Fas/FasL system plays an important role in the apoptosis regulation. ”

Line 184, “FasL expressed by cytotoxic T lymphocytes combing with Fas expressed by target cells, results in the formation of Fas-associated death domain (FADD)” is changed to “When external FasL is expressed by cytotoxic T lymphocytes and combined with Fas which is expressed by target cells, Fas-associated death domain (FADD) will be formed”.

Line 187, “ induced apoptosis of human colon cancer HT-29 cells via DR3 pathway” is changed to “induced HT-29 colon cancer cells apoptosis”.

Line 188, “DR3, one of the death receptors, activate apoptosis through TRADD, FADD and caspase-8” is changed to “Caspases is a family of cysteine proteases and act on proteins or enzymes related to cytoskeleton or DNA, responsible for apoptosis. DR3 activated apoptosis through triggering TRADD, FADD and caspase-8”

Line 191, “cordycepin induced apoptosis human prostate carcinoma LNCaP cells” is changed to “cordycepin induced apoptosis in human prostate carcinoma LNCaP cells”.

Line 193, “down-regulating expression of anti-apoptotic Bcl-2 and up-regulating pro-apoptotic Bax” is changed to “and causing a a dose-dependent increase of pro-apoptotic Bax and decrease of anti-apoptotic Bcl-2”.

Lline 196, “treatment” is deleted, and “level” is changed to “levels”.

Line 198, “the extract of cultured C. militaris” is changed to “C. militaris extract”.

Line 200, “, and decreasing the protein expression of X-linked inhibitor of apoptosis protein (XIAP) through increasing” is changed to “and”.

Line 203, “A3” is changed to “A3”, and “on the B16-BL6 cell membrane” is changed to “on the B16 cell membrane”.

Line 208, “whereas depletion of A3R abrogated the effect of cordycepin” is changed to “since the apoptosis-inducing effect of cordycepin is abolished with depletion of adenosine receptors”

Line 209, “ And adenosine A2A receptor (A2AR) antagonist and small interference RNA (siRNA) knockdown of A2AR blocked cordycepin-induced apoptosis in C6 glioma cells [129]“ is deleted.

Line 213, “Ma et al. [16] identified a novel apoptosis mechanism that adenosine induces apoptosis by increasing reactive oxygen species (ROS) level to mediate mitochondrial membrane dysfunction in BEL-7404 liver cancer cells“ is changed to “ Ma et al. [16] found a novel apoptosis mechanism that extracellular adenosine could trigger apoptosis by increasing reactive oxygen species (ROS) production and mitochondrial membrane dysfunction in BEL-7404 liver cancer cells.”

Line 218, “both” is added behind “via”.

Line 222, “human colon cancer HCT116 cells“ is changed to “ HCT116 human colon cancer cells”.

Line 223, “HCT116 cells” is deleted.

Line 224, “suppressed” is changed to “inhibited”.

Line 226, “CSP decreased PI3K protein expression and phosphorylation level of AKT and mTOR (mechanistic target of rapamycin), and increased AMPK (adenosine 5‘-monophosphate-activated protein kinase) protein expression and phosphorylation level of ULK1 (Unc-51-like kinase1), activating AMPK/mTOR/ULK1 pathway and suppressing PI3K/Akt/mTOR pathway” is changed to “signal pathways of PI3K-AKT-mTOR and AMPK-mTOR-ULK1 were all involved.”

Line 232, “accompanied by autophagosome marker LC3-II levels increased” is changed to “accompanied with the increase of autophagosome marker LC3-II levels.”

Line 237, “Cyclin proteins interact with their corresponding cyclin-dependent kinases (CDKs) to form cyclin/CDK complexes which regulate cell cycle processes.” is deleted.

Line 238, “G1, S and G2/M phases are crucial checkpoints in the cell cycle processes. Arresting the cell cycle in the G1, S and G2/ M phases can lead to the inhibition of tumor cells proliferation and induction of apoptosis” is changed to “The cell cycle can be mainly divided into four phases, of which G1, S and G2/M phases are crucial checkpoints in the cell cycle processes. Cell cycle arrest in the G1, S and G2/ M phases can lead to the inhibition of tumor cells proliferation and induction of apoptosis”.

Line 241, “to arrest cell cycle” is changed to “to cause cell cycle arrest”.

Line 242, “Cordycepin resulted in growth inhibition of bladder cancer 5637 and T-24 cells and human colon cancer HCT116 cells, through G2/M phase cell cycle arrest by up-regulating the expression of p21WAF1 (a universal key inhibitor in regulating cell-cycle progression) and down-regulating expression of G2/M cell cycle regulatory proteins (cyclin B1, Cdc25c and Cdc2) through JNK1 signal pathway” is changed to “Cordycepin inhibited growth of 5637 and T-24 bladder cancer cells and HCT116 colon cancer cells, through G2/M cell-cycle arrest. The expression of p21WAF1 (a universal key inhibitor in regulating cell-cycle progression) was up regulated and cyclin B1, Cdc25c and Cdc2 (G2/M cell-cycle regulatory proteins) were down regulated, through JNK1 signal pathway”.

Line 247, “of the cell cycle” is deleted.

Line 248, “decrease of G1 and G2/M cell numbers, and decrease of cell viability” is changed to “the decrease of G1 and G2/M cell numbers and cell viability”.

Line 253, “Because of its similar structure with adenosine, cordycepin (3′-deoxyadenosine) can participate in the synthesis of DNA or RNA for many enzymes fail to distinguish between adenosine and 3′-deoxyadenosine, which results in a failure further incorporation of nitrogenous bases (A, U, G and C) and the premature termination of transcription” is changed to “In addition, cordycepin is a transcription and polyadenylation inhibitor, and affecting the RNA synthesis. A study showed that cordycepin caused accumulation of the corresponding triphosphate derivative, 3′dATP, which might lead to the incorporation of analogue into nascent nucleic acid oligonucleotides and RNA synthesis inhibition”.

Line 264, The paragraphMetastasis is one of the primary causes of mortality in cancer patients, and is the end result of multiple interactions including invasion between tumor and host. Epithelial-mesenchymal transition (EMT) plays roles in promoting cancer cells invasion and metastasis. The elevated expression of EMT-related proteins such as matrix metalloproteinases (MMPs), especially MMP-2 and MMP-9, leads to the degradation of extracellular matrix (ECM) components and tissue invasion [134, 135]. Many studies have shown that MMP expression is correlated with tumor progress and metastasis [135, 136]. A study showed that cordycepin mediated antitumor invasion through decreasing NF-κB / MMP-9 pathway in TNF-α- induced invasion and migration in bladder cancer 5637 and T-24 cells were inhibited by cordycepin through decreasing the promoter activity of the MMP-9 gene and MMP-9 expression [104]. Further investigation showed that cordycepin reduced the transcriptional activity of the transcription factors, NF-κB and activator protein-1 (AP-1), which were identified as by gel-shift assay as cis-elements for TNF-α activation of the MMP-9 promoter. In addition, a novel water‐soluble polysaccharide CME‐1 isolated from O. sinensis mycelia, was found to inhibit migration of B16-F10 melanoma cells, and the mechanism was that CME-1 reduced MMP-1 expression and down-regulated the phosphorylation level of ERK1/2 and p38 MAPK [59]. The recent researches have uncovered the mechanisms that Chinese Cordyceps induces tumor cells death, cell cycle arrest and inhibits migration, as shown in Table 4. Figure 3 gives a graphic summary of antitumor effect and mechanism by Chinese Cordyceps.” is replaced by “Metastasis referring to the movement of cancer cells from primary tumor sites to other organs and tissues, is the end result of multiple interactions including invasion between tumor and host, indicating uncontrolled spread of the tumor cells.  Epithelial-mesenchymal transition (EMT) related proteins such as matrix metalloproteinases (MMPs) play an important role in metastasis. For example, MMP-2 and MMP-9 can lead to the degradation of extracellular matrix (ECM) components and tissue invasion [134- 136]. A study showed that cordycepin inhibited 5637 and T-24 cells invasion through decreasing MMP-9 expression and he transcriptional activity of activator protein-1 (AP-1), which were identified as by gel-shift assay as cis-elements for TNF-α activation of the MMP-9 promoter via NF-κB / MMP-9 pathway [104]. In addition, a novel polysaccharide CME‐1 isolated from O. sinensis was found to inhibit migration of B16-F10 melanoma cells, and the mechanism was that CME-1 reduced MMP-1 expression and down-regulated the phosphorylation level of ERK1/2 and p38 MAPK [59].”

  1. Reviewer: For the conclusion section, too many details were put herewith. The authors need to summarize the vital point of the work only. The minor point may be moved to the sub-topic in each part for discussion or related to that section.

Responses: “Chinese Cordyceps has been recognized as notable traditional Chinese medicine, possessing immunomodulatory, anti-inflammatory and anti-tumor effects. Scientific research shows that the regulation of immunity system by traditional Chinese medicine could be one of the therapeutic strategies for cancer treatment. Adenosine, cordycepin and polysaccharides are major bioactive components of Chinese Cordyceps and have anti-tumor and immunomodulatory properties, since they not only influence the growth of tumor cells but also activate immune cells in TIM. The formation and dynamic changes of TIM are a multifaceted process and involved in various types of immune cells and signal pathways. Here we maintain both extracts of Chinese Cordyceps and specific compounds (especially adenosine, cordycepin and polysaccharides) have a certain effect on regulating the expression of cytokines in TIM. Chinese Cordyceps plays an important role in enhancing tumor cells antigenic responses, and at the same time, it reverses suppressive phenotype of immune cells to suppresse tumorigenesis, which promotes its antitumor functions. Thus, it is of significance to promote proinflammatory phenotype of immune cells and remodel suppressive phenotype to improve their antitumor activity. More importantly, Chinese Cordyceps has been found to have notable direct anti-tumor and anti-metastatic activities in vivo and in vitro. It can inhibit the growth of tumor and proliferation of tumor cells by inducing both apoptosis and autophagy, as well as preventing or overcoming metastasis. Many signaling pathways including MAPKs, NF-κB, PI3K/Akt, Caspase and AR participate in the anti-tumor process. Nowadays, the anti-cancer ability of Chinese Cordyceps has been a subject of research for approximately 60 years, for cancer is one most complicated disease with diverse pathogenic mechanism. Efforts from more aspects can be taken and combining Chinese Cordyceps to treat cancer may provide a potential therapeutic strategy.” is replaced by “In conclusion, Chinese Cordyceps has significant antitumor activity and immunomodulatory activity. On the one hand, it can directly act on tumor cells to kill tumor cells or inhibit tumor growth and effectively attenuate tumor cell metastasis. On the other hand, Chinese Cordyceps can change the tumor microenvironment and enhance antitumor immune responses by down-regulating the expression of immunosuppressive factors and up-regulating the expression of pro-inflammatory factors, thereby improving the antitumor function. These findings may provide therapeutic strategies for treating cancer.” is added.

In addition, we have checked and polished the English language carefully, while we invite a native English speaker to smooth the whole manuscript.

I hope this presentation will make you feel more satisfactory. I am looking forward to hearing from you soon. Thank you for your attention.

Reviewer 2 Report

The manuscript describes the structures of the main component of Chinese cordyceps, and further discussed the underlying mechanism pertaining to its antitumor and immunomodulatory properties. While the review is interesting and has merit, some improvements are required prior to consideration for acceptance.

Various similar reviews have been published on Chinese cordyceps with regard to their structural characteristics and antitumor properties. While the current review is still relevant to the scientific community, improvements are needed to strengthen the aspect of manuscript novelty.

Line 70 mentions on physicochemical properties, however discussions on physicochemical properties seem lacking.

Suggest discussing and including the preparation/ method of bioactive extraction/ type of extracts in Table 1 to strengthen the manuscript content.

As angiogenesis is a vital process for tumor growth, it would also be interesting to include in detail the effects of Chinese cordyceps on angiogenesis as well as the mechanistic basis underlying its antiangiogenic properties.

Table 4 should comprehensively summarize various other models that have been widely utilized to investigate antitumor properties, i.e. lung carcinoma, gastric adenocarcinoma, renal, cervical and oral cancers, among others.

It is also recommended that the authors add some discussion on future trends and a novel perspective regarding the utilization of Chinese cordyceps in the aspect of immunopharmacology.

The latter part of the conclusion is not coherent with the content of the manuscript as the authors are emphasizing on immunomodulation in Covid19. Since the manuscript does not discuss anything pertaining to infectious diseases, the conclusion pertaining to Covid19 should be omitted, and instead should be focusing on immunomodulation of antitumor activity by Chinese cordyceps.

Overall, the language including the abstract needs improvement and should be checked meticulously. For example: Line 9-12, Line 21-22, Line 61, Line 122, Line 132.

Author Response

Response to Reviewer 2 Comments 

  1. Reviewer: Line 70 mentions on physicochemical properties, however discussions on physicochemical properties seem lacking. Suggest discussing and including the preparation/ method of bioactive extraction/ type of extracts in Table 1 to strengthen the manuscript content.

Responses: We sorted out the bioactive substances and their functions in O. sinensis, and found three components with significant anti-tumor and immunomodulatory activities, adenosine, cordycepin and polysaccharide. Therefore, this review focuses on the antitumor activities and mechanisms of these three active substances. In view of the reviewers' comments, we have added the preparation and extraction methods in 2.1 Adenosine and Cordycepin and 2.2 Polysaccharides, respectively.

  1. Reviewer: As angiogenesis is a vital process for tumor growth, it would also be interesting to include in detail the effects of Chinese cordyceps on angiogenesis as well as the mechanistic basis underlying its antiangiogenic properties.

Responses: “Angiogenesis is vital for organ growth and repair, and essential for the tumor growth. The Vascular endothelial growth factor (VEGF) family plays an important role in angiogenesis. VEGF, a key angiogenic growth factor, has higher expression level in tumor tissues and can accelerate the differentiation, proliferation, and migration of endothelial. Chinese Cordyceps has been demonstrated to inhibit the VEGF/VEGFR2 signaling pathway and exert antiangiogenesis function [126]. Besides, the Overexpression of proto-oncogenes c-Myc and c-Fos may promote tumor cells proliferation under growth promoting stimulation. c-Myc, encoding a ubiquitous transcription factor and promoting cell division, is related to apoptosis and the occurrence and development of various tumors. c-Fos, essential for cell proliferation, can up regulate cell cycle by induction of cyclin D1 [151]. c-Fos is expressed at low level in normal cells while overexpressed in tumor cells. Yang et al. [130] found that EPSF isolated from C.sinensis could down regulate the expression of VEGF, c-Myc and c-Fos, which was the important factor to inhibit tumor growth, invasion, and metastasis.” is added behind line 278.

  1. Reviewer: Table 4 should comprehensively summarize various other models that have been widely utilized to investigate antitumor properties, i.e. lung carcinoma, gastric adenocarcinoma, renal, cervical and oral cancers, among others.

Response: I agree with reviewer’s suggestion, and Table 4 has been supplemented with several cancer types, such as gastric, cervical, and oral cancer, and the table has been adjusted by cancer type.

  1. Reviewer: It is also recommended that the authors add some discussion on future trends and a novel perspective regarding the utilization of Chinese cordyceps in the aspect of immunopharmacology.

Responses: I agree with reviewer’s suggestion, and have added some discussion in “4. Discussion”, as follows: “In recent years, the traditional therapy for cancer has become an attention direction of researchers, and many researchers believe that traditional therapy is a potential new therapy. The pathogenesis of cancer is diverse and complex, and Chinese Cordyceps has many active ingredients and diverse extracts, which can inhibit the growth of various tumors, prevent or overcome metastasis through various pathways (Figure 3). It is well known that improving self immunity can lay a good foundation for fighting and treating many diseases. Chinese Cordyceps has a long history of use in China, and much evidence suggests that Chinese Cordyceps, acting as an immune response activator, is used for the treatment of a variety of diseases including cancer. Increasing studies have shown that Chinese Cordyceps has immunomodulatory, anti-inflammatory, and antioxidant activities that affect the immune system and TME in various ways. The polarization and remodeling of the phenotype of immune cells (such as T cells and macrophages) by Chinese Cordyceps have effects on cytokines production in TME, which may affect tumor progression. The anticancer ability of Chinese Cordyceps has been the subject of research for nearly 60 years and its anti-tumor effect has been confirmed in cancer cells or mouse cancer models alone or in combination with other drugs. The research on clinical application still needs more efforts.”

  1. Reviewer: The latter part of the conclusion is not coherent with the content of the manuscript as the authors are emphasizing on immunomodulation in Covid19. Since the manuscript does not discuss anything pertaining to infectious diseases, the conclusion pertaining to Covid19 should be omitted, and instead should be focusing on immunomodulation of antitumor activity by Chinese cordyceps.

Response: “With the development of immunology, the regulation on TIM of Chinese Cordyceps will play a vital role in treatment of cancer and immune disorders, and function as an important adjuvant immunotherapy candidate for disease therapy and health regulation. The current novel coronavirus disease 2019 (COVID-19) outbreak, caused by severe acute respiratory syndrome coronavirus 2 (SARS-CoV-2), resulted in a rapid increase in infected patients and high mortality rate all over the world. Research shows that SARS-CoV-2 leads to uncontrolled inflammatory responses characterized by significant pro-inflammatory cytokine release [139], lymphopenia and lymphocyte dysfunction [140] in COVID-19 infected patients, which may lead to immune abnormalities and severe multiple organ dysfunction [141] in patients with COVID-19. The immune system is the best defense, and it is vital to enhance the management of SARS-CoV-2 response. In view of performance in immune regulation, it can be expected that Chinese Cordyceps can play a role in improving antiviral immunity since it has over 20 bioactive ingredients playing a synergistic role, which may aid the development of new therapeutic strategies against COVID-19 in clinical trials. ” is deleted.

  1. Reviewer 3: Overall, the language including the abstract needs improvement and should be checked meticulously. For example: Line 9-12, Line 21-22, Line 61, Line 122, Line 132.

Response: page1 line9, “The special growth environment and unique growth mode make Chinese Cordyceps rich in various active components, adenosine, cordycepin and polysaccharides are recognized as the main components, and have been proved with multiple pharmacological activities, especially on with significant immunomodulatory and antitumor effects.” is changed to “It is rich in various active components, of which, adenosine, cordycepin and polysaccharides have been confirmed with significant immunomodulatory and antitumor functions”.

Page 1 line 13, “the underlying mechanism is poor understood” is changed to “the underlying mechanism of antitumor remains poor understood”.

Page 1 line 15, “Analysis found that antitumor effects is” is changed to “Analysis found that Chinese Cordyceps promote immune cells antitumor function by”.

Page 1 line 17, “Besides, directly inhibiting the growth of tumor cells is by inducing apoptosis and autophagy, cell cycle arrest, and inhibition of migration, invasion and metastasis” is changed to “Moreover, Chinese Cordyceps can inhibit growth and metastasis of tumor cells by death (including apoptosis and autophagy) induction, cell cycle arrest and angiogenesis inhibition”.

Page 1 line 19-23, “Recent evidence has revealed the molecular pathways involving mitogen-activated protein kinases (MAPKs), nuclear factor kappaB (NF-κB), cysteine-aspartic proteases (caspases) and serine/threonine kinase Akt” is changed to “Recent evidence has revealed the signal pathways of mitogen-activated protein kinases (MAPKs), nuclear factor kappaB (NF-κB), cysteine-aspartic proteases (caspases) and serine/threonine kinase Akt were involved in the antitumor mechanisms. Furthermore, the pathways are mediated by the pathways are mediated by putative receptors, such as adenosine receptors (AR) and death receptors (DRs)”

Line 61, “pharmology” is changed to “pharmacological”.

Line 120-124, “Chinese Cordyceps polysaccharides include extracellular polysaccharide (EPS) and intracellular polysaccharide (IPS). EPS resource from the fermentation broth of submerged Chinses Cordyceps fungi, while IPSs resource from fruiting bodies of Chinese Cordyceps and cultured mycelium” is changed to “Cordyceps polysaccharides include two forms: extracellular polysaccharide (EPS), mainly resourcing from the fermentation broth of submerged Cordyceps spp., and intracellular polysaccharide (IPS), mainly resourcing from the fruiting bodies of Chinses Cordyceps and cultured mycelium”.

Line 132, “Scheme 109.” is changed to “Studies have shown that the occurrence and development of tumors are closely related to immune surveillance. Immunotherapy has been proved to be an effective method to treat a variety of cancers [109],”.

In addition, we have checked and polished the English language carefully, while we invite a native English speaker to smooth the whole manuscript.

I hope this presentation will make you feel more satisfactory. I am looking forward to hearing from you soon. Thank you for your attention.

Round 2

Reviewer 2 Report

Dear authors,

The majority of the remarks have been addressed. However, I would suggest changing the section "4. Discussion" to Future insights or perspectives for better representation. 

Errors e.g.: Table 3. "supernatan", "Live cancer"